# Prerequisite Binding Modes Determine the Dynamics of Action of Covalent Agonists of Ion Channel TRPA1

**DOI:** 10.3390/ph14100988

**Published:** 2021-09-28

**Authors:** Balázs Zoltán Zsidó, Rita Börzsei, Erika Pintér, Csaba Hetényi

**Affiliations:** 1Department of Pharmacology and Pharmacotherapy, Medical School, University of Pécs, Szigeti út 12, 7624 Pécs, Hungary; zsido.balazs@pte.hu (B.Z.Z.); erika.pinter@aok.pte.hu (E.P.); 2Department of Pharmacology, Faculty of Pharmacy, University of Pécs, Szigeti út 12, 7624 Pécs, Hungary; rita.borzsei@gmail.com

**Keywords:** TRPA1 receptor, prerequisite binding, covalent binding

## Abstract

Transient receptor potential ankyrin 1 (TRPA1) is a transmembrane protein channeling the influx of calcium ions. As a polymodal nocisensor, TRPA1 can be activated by thermal, mechanical stimuli and a wide range of chemically damaging molecules including small volatile environmental toxicants and endogenous algogenic lipids. After activation by such compounds, the ion channel opens up, its central pore widens allowing calcium influx into the cytosol inducing signal transduction pathways. Afterwards, the calcium influx desensitizes irritant evoked responses and results in an inactive state of the ion channel. Recent experimental determination of structures of apo and holo forms of TRPA1 opened the way towards the design of new agonists, which can activate the ion channel. The present study is aimed at the elucidation of binding dynamics of agonists using experimental structures of TRPA1-agonist complexes at the atomic level applying molecular docking and dynamics methods accounting for covalent and non-covalent interactions. Following a test of docking methods focused on the final, holo structures, prerequisite binding modes were detected involving the apo forms. It was shown how reversible interactions with prerequisite binding sites contribute to structural changes of TRPA1 leading to covalent bonding of agonists. The proposed dynamics of action allowed a mechanism-based forecast of new, druggable binding sites of potent agonists.

## 1. Introduction

Mammalian neurons of the pain pathway detect potentially dangerous environmental signals. In the peripheral nervous system, there are specialized nociceptive neurons, that recognize either noxious chemical signals, thermal or mechanical stimuli. Pathological processes, such as tissue damage and inflammation elicit the formation and subsequent release of a wide variety of mediators (arachidonic acid derivatives, free radicals, H_2_O_2_, H_2_S, etc.). These molecules depolarize the nerve terminals of nociceptors, which transmit the signals to the central nervous system. The transient receptor potential ankyrin 1 (TRPA1) is a Ca^2+^-permeable cation channel that was identified as the chemical nocisensor, expressed by primary afferent nerve fibers [1,2,3,4,5,6,7,8,9]. Activated TRPA1 promotes pain itching and induces local neurogenic inflammatory response via the release of neuropeptides, such as substance P, calcitonin gene-related peptide and neurokinins.

TRPA1 receptor is activated by the binding of electrophile ligands (Figure 1) to its N-terminus cytoplasmic binding site (Figure 2A), which is characterized by three nucleophilic cysteine residues (C621, C641 and C665) [6,7]. This binding event induces a local conformational change, that is translated to the whole of the receptor, and a 15° rotation of the transmembrane domain is observed [7], resulting in a pore widening, that facilitates Ca^2+^ influx, which first potentiates, then desensitizes agonist-induced responses [7], resulting in an inactive state of the TRPA1 ion channel [10]. On the local scale, a cytoplasmic A-loop near the transmembrane region of the receptor overlays the binding site cavity [7] and initially sterically hinders agonist binding. Upon the binding of the electrophile agent, a flip of the A-loop (residues 666–680, Figure 2B) was observed towards the cell membrane, leaving more binding space for the agonists. Thus, flipping of A-loop contributes to the widening of the pore of the ion channel, and the above-mentioned activation process [7] at the same time, and therefore, it is important in agonist design.

Known electrophile agonists of the TRPA1 receptor include dimethyl trisulfide (DMTS) and allyl isothiocyanate (AITC) [4,5,8]. However, given the size of these molecules, they would not bind selectively to only one cysteine amino acid residue in the body. Thus, the need for a selective site-specific electrophile agonist (JT010) was first met in 2015 by Takaya et al. [4], and a potent thiazol derivate agonist (EC_50_ = 0.65 nM, Figure 1) with a covalent alkyl-halide warhead was designed. Since then, the binding position of JT010 was experimentally found by cryo-electron microscopy [7]. Recent structural studies [2,3,6,7] provided additional details of the agonist binding mechanism and consequential receptor activation. Besides JT010, the binding of the other two site-specific covalent agonists BITC [6] and bodipy-iodoacetamide [7] (Figure 1) was investigated, which preferentially bind to the active site cysteine C621 of the TRPA1 receptor.

Covalently binding agonists are subjects of intense research [11], and they often form covalent bonds with nucleophilic cysteine residues. As cysteine is abundant in the human proteome, a careful design has to be performed to achieve binding site specificity [11,12] to avoid unwanted promiscuity of the agonists via non-selective covalent bonds with non-targeted cysteines. Thus, a common strategy of covalent agonist design adopts known agonists having selective non-covalent interactions [11] with the receptor. Both covalent and non-covalent interactions have been fully described [6,7] at the final, irreversible binding mode of the agonists in Figure 1. However, the binding routes leading to the final binding pocket have not been mapped at atomic resolution. The above-mentioned, agonist-induced structural changes of TRPA1 at the A-loop (Figure 2B) suggest that the agonist may form dynamic interactions with prerequisite sites on their route to the final, covalent positions.

The present study is focused on the elucidation of binding dynamics of covalent agonists using experimental structures of their complexes with TRPA1 (holo form) and compared to prerequisite interactions with the apo form. Covalent and non-covalent molecular docking techniques are tested at forecasting prerequisite and final, covalent binding modes of the agonists. The docked agonist-TRPA1 complexes are subjected to molecular dynamics calculations to complete the binding mechanism at the atomic level. We also aim at the mechanism-based forecasting of prerequisite binding sites which can become druggable targets of agonists in drug design projects.

## 2. Results and Discussion

### 2.1. Final Covalent Binding Modes

The atomic resolution structures of three covalent agonists JT010 [4], BITC [6] and bodipy-iodoacetamide [7] (Figure 1) bound to the TRPA1 ion channel are available in the Protein Databank (PDB). JT010 and bodipy-iodoacetamide have an alkyl-halide covalent warhead, the bonds made with chlorine and iodine atoms break up during covalent binding to C621 (Figure 3). BITC participates in an isothiocyanate bond with C621 (Figure 3), the N=C double bond diminishes, and the C atom forms a covalent bond with the S atom of the target C621, and the N atom of BITC gains a H atom. As the nucleophilic C621 binds all three electrophilic agonists covalently [6,7], our docking studies focused on the surrounding binding pocket.

As a first step of our investigation, the popular program package FITTED [13,14,15] was tested using the experimental PDB structures as references for comparison with the docking results. FITTED was first tested to reproduce the final binding modes of the covalently bound agonists. A standard evaluation protocol was applied to all covalent docking results. Firstly, the structural match of the calculated binding mode (bind position, orientation, and conformation of the ligand) to the crystallographic reference was calculated and the best match was expressed as a root mean squared deviation (*RMSD*_best_, see Section 4 for the definition of all metrics used in the Tables) [16]. Secondly, it was tested if the docking method identified *RMSD*_best_ as an energetically favorable binding mode and ranked it at the top of the list of all binding modes. The second criterion reports on the applicability of the binding free energy (scoring) function of the docking method.

The covalent docking of the agonists to the holo form of TRPA1 was structurally successful as the *RMSD*_best_ values were comparable/below 2.5 Å (Table 1), a threshold accepted in the literature [17,18,19,20,21]. Only bodipy-iodoacetamide showed a slightly elevated *RMSD* (Table 1), which is due to the mobility of the -SH group around the C_β_ of C621 during docking. The rotation of the S atom around the C_β_ of C621 also turns bodipy-iodoacetamide from its experimental position around its longitudinal axis by ca. 180°. The C_β_-S-C_bodipy-iodoacetamide_ angle is also smaller than that observed in the experimental structure. The binding modes with *RMSD*_best_ were positioned to the first place on the ranking lists for all three agonists (Table 1). The calculated free energy of binding of JT010 is the most favorable, closely followed by that of BITC (Table 1), apparently, the alkyl-halide covalent bond formed by the alkyl-iodine pharmacophore of bodipy-iodoacetamide is less favorable compared to that formed by the alkyl-chlorine pharmacophore of JT010. The efficiency index (EI_NHA_) of BITC is the best out of the three reference ligands.

The large conformational flexibility of target molecules is a challenge for fast docking programs and can be handled by the involvement of molecular dynamics approaches [22] which require longer calculation times. Conformational flexibility is important when induced fit occurs during agonist binding. Here, the A-loop is a flexible element that covers the binding site (Introduction, Figure 2B) in the apo TRPA1 conformation, and sterically prevents the agonist from reaching its destination (holo position) at the bottom of the pocket resulting in unacceptably large *RMSD* values and positive energies (Appendix A). Therefore, the A-loop was removed from TRPA1 and covalent docking calculations were repeated using FITTED. Although the A-loop consists of the amino acids from 660–680, in the apo docking calculations, only the amino acids 665–677 were removed, as these are the ones that elicit the greatest movement during apo to holo transition. In the absence (Table 1, Table 2, Table 3) of the A-loop, all covalent bonds were formed with the apo TRPA1 (Table 1), with *RMSD* values larger than those observed in the case of the holo TRPA1 (Table 1). The corresponding ΔG_best_ values were lower than those of the apo TRPA1 by 6% on average. These findings emphasize the role of A-loop-agonist interactions in the final binding position. However, the removal of the A-loop did not influence the EI and ΔG_best_ order of the ligands.

The formation of the covalent bond between the agonist and TRPA1 (Figure 3) is a quantum mechanical phenomenon, which is hard to treat adequately by docking programs based on molecular mechanics scoring functions [23,24]. Despite the above challenges, the covalent docking methodology of FITTED performed well for the above test cases and supplied relevant structural and scoring (ranking) results. Encouraged by the above test results, we expect that FITTED will also help in mapping the prerequisite binding modes on route to the final binding pocket.

### 2.2. Prerequisite Binding Modes

As it was discussed in Section 1, the entrance to the binding cavity (outer prerequisite binding mode) and the formation of the final ligand-target covalent bond (inner prerequisite binding mode) is hindered by the position of the A-loop (Figure 2B) in the apo form of the TRPA1 target. During a successful binding process, the ligand initiates the flipping of the A-loop via intermolecular interactions with the loop. To develop such interactions, the ligand needs to occupy a prerequisite binding mode outside the final binding pocket. Two different programs, FITTED and AutoDock 4.2.6 (The Scripps Research Institute, La Jolla, CA, USA) [25], were involved in the mapping of possible prerequisite binding modes by non-covalent docking and the results are shown in Table 2 and Table 3, respectively. Both programs are based on physico-chemical principles. FITTED is a genetic algorithm-based docking method, that includes an ESFF [26] force field-based search engine, called CDiscoVer [27] to perform conjugate gradient minimizations [13]. AutoDock also uses a (Lamarckian) genetic algorithm and AMBER-based intermolecular force field terms for scoring [25].

For prerequisite binding modes *RMSD* was not calculated, the distance (d) between TRPA1 C621 S atom and the atom of the agonist that participates in the covalent binding was used as a measure of ligand position instead. The d_best_ value indicates the closest distance between the aforementioned atoms (black dots in Figure 3) achieved by subsequent docking calculations. The match of the docked binding mode was expressed as the percentage of matching amino acids (AA_match_) compared with the experimental binding pocket. Both programs ranked the binding mode of BITC with the d_best_ as the top 1st in all prerequisite docking calculations (Table 2 and Table 3). However, BITC is considerably smaller, than JT010 and bodipy-iodoacetamide. The head-to-tail docking orientation of larger agonists, like JT010 and bodipy-iodoacetamide might cause elevated d_best_ values.

In all cases and both scorings, the holo prerequisite docking calculations yielded better AA_match_ and Δ*G_FD_* and Δ*G_AD_*, than the apo prerequisite docking calculations (Table 2 and Table 3). This was expected, as the holo conformation of the binding site is already prepared to accept the agonists. The FITTED prerequisite holo docking calculations yielded an AA_match_ of 100% in the cases of all three agonists. The d_best_ values of the prerequisite docking calculations were under 4.0 Å in all cases, with the only exception of bodipy-iodoacetamide holo docking (Table 3). In the case of AutoDock, the d_best_ values of the holo prerequisite calculations were below 7.0 Å, and slightly above it in the apo prerequisite docking runs. The d_best_ ≥ 7.0 Å values are due to head-to-tail binding mode of the agonists (Table 2 and Table 3).

The comparison of the prerequisite binding modes produced by FITTED on both the holo and apo docking calculations of the three compounds resulted in C621 as the only common binding site amino acid for all three agonists. All holo prerequisite docking calculations found C621 with only the exception of bodipy-iodoacetamide prerequisite holo docking with AutoDock. These findings suggest that different prerequisite binding modes (Appendix A) might result in good final covalently binding positions (Table 1). By investigating more agonists with two docking programs, one might expect to discover common amino acids that indicate a larger prerequisite binding area. If an agonist at least partially interacts with the prerequisite binding area it has a chance to find its way to the final binding pocket.

Amino acids C665, P666 and F669 of the A-loop are part of the binding pocket of bodipy-iodoacetamide, and also of most prerequisite binding modes of bodipy-iodoacetamide and JT010 (Appendix A) found by both programs. C665 is also highlighted in the literature [7] as an important amino acid both in agonist binding and receptor activation. The absence of interaction of BITC with the above-mentioned amino acids might be due to the smaller size of BITC compared to the other two agonists (Figure 1). These results suggest a previously unexplored structural role of the amino acids P666 and F669 co-operating with C665 in agonist binding and consequent flipping of the A-loop, leading to conformational changes and receptor activation. These findings were also strengthened by virtual mutation and docking (Appendix A). However, BITC interacts with another part of the A-loop as prerequisite binding site found by apo docking with AutoDock. This BITC site was also sufficient to open the binding pocket in molecular dynamics (MD) simulations (Section 2.3).

Although not an accepted medicine, JT010 has a remarkable EC_50_ of 0.65 nM [4]. Thus, in novel drug design, JT010 can be regarded as a reference point, and therefore, it was further investigated from the mechanism viewpoint of this Section. During the transition from a non-covalent, prerequisite binding mode (d = 3.6 Å) to the final, covalent binding mode of JT010, its interactions with F669 and Y680 diminish (Appendix A), and new interactions with K620, I623 and E625 are formed (cut-off distance of interaction of 3.5 Å for heavy atom–heavy atom distance). Interactions with the two binding site cysteines, C621 and C665 [7] are observed for both non-covalent and covalent binding modes of JT010. As it was highlighted in the previous section, F669 is part of the A-loop and has a possible role in the flipping of the loop during agonist binding to the TRPA1 receptor, based on the example of JT010. It can be hypothesized, that interaction with F669 is only important in the initial prerequisite binding of the agonist, later during covalent bond formation this interaction diminishes, and the agonist penetrates deeper into the binding site, interacting with amino acid residues that are in close proximity of C621. This observation is strengthened by MD simulations also (see Section 2.3). The covalent Δ*G_FD_* of JT010 almost doubles (and consequently its EI also), compared to that of the prerequisite binding mode.

Regardless of their potency, all three agonists can activate the TRPA1 ion channel, however, to prevent xenobiotic overload of the body it is advisable to administer the lowest possible dose of a drug, which is only effective if the agent is highly potent. If using JT010 as a reference for future studies, the following limit values can be concluded for the selection of potent agonists. A prerequisite EI value of at least 2 kcal/mol, and a prerequisite d_best_ ≤ 4.0 Å, and a prerequisite Δ*G_FD_* of at least -35 kcal/mol forecast a strong agonist. Notably, the Δ*G_FD_* value of JT010 is approximated by that of BITC, and the EI of BITC even surpasses that of JT010 (Table 1). bodipy-iodoacetamide somewhat lags behind JT010 and BITC. These findings are in good agreement with the literature, as the EC_50_ value of iodoacetamide (without the bodipy label) is 357 µM [28], which is substantially larger, than that of JT010. The EC_50_ of allyl-isothiocyanate (a similar compound to BITC) is 37 nM [7], which is also in the nanomolar range, as the EC_50_ of JT010.

The docking performance of FITTED slightly outperformed AutoDock as seen in Table 2 and Table 3. However, FITTED requires a probe previously placed within the binding site to select the binding site amino acids, which obviously helps the search. At the same time, AutoDock did not require such information, and an unrestricted search could be performed for the prerequisite binding mode within the docking box. Thus, we decided to use the prerequisite binding mode of BITC found by AutoDock apo calculation with A-loop (Appendix A) for further MD simulations in the next, Section 3.

### 2.3. Ligand Migration Dynamics Connecting Prerequisite and Final Binding Modes

MD simulations (100 ns, unrestrained, with explicit waters and simulated annealing protocol as described in Section 4) were performed on both apo and holo forms of TRPA1 (Table 4) to further explore the binding dynamics of the agonist BITC of the best EI value (Table 1). As the results of the previous Section indicated that the prerequisite binding modes affect A-loop, we were particularly interested in the structural changes of the loop, and the communication between the distinct prerequisite and final binding modes. An MD_apo_ simulation was used as a reference, to observe if there are any changes in the conformation of the A-loop in the absence of the agonist. Then, two MD simulations were started from two prerequisite binding modes of BITC on the apo TRPA1 found in the previous Section, one of them interacting with the loop (MD_rank1_). Finally, an MD simulation was started from the experimental binding position of BITC (MD_holo_), with the covalent bond cut and the geometry of the N=C=S bonds restored. In the MD_holo_ simulation, an unbinding-binding event occurred and the A-loop remained stable throughout the simulation (Figure 4). The interaction with the original five amino acids of the TRPA1 pocket gradually diminished and appeared once again in a very short time interval of the first 0.7 ns (Figure 4). During the entire MD_apo_ (Table 4) simulation, no significant changes were observed in the conformation of the A-loop, while, in the case of MD_rank1_, the loop moved upward (the red and blue arrows show the movement of A-loop and the teal arrows the movement of BITC on Figure 5). In the starting position of MD_rank1_, BITC interacted with the loop and was positioned beneath it (marked with 0 ns at Figure 5). After a very short time (0.3 ns) the ligand dissociates from the TRPA1 surface, dragging down the loop with itself. After 17.6 ns the loop moved upwards, approximating the open position of the binding site which is present in the holo structure (Figure 2 and Figure 5). Finally, (at 38 ns) BITC finds its way back into the binding pocket and resides there for 2 ns until dissociation. At the same time, in the case of MD_rank3_ (Table 4) in which the docked position of BITC did not interact with the A-loop, BITC dissociated after 1.6 ns and afterwards, no changes were observed in the structure of the loop. Thus, the above results showed how the binding of an agonist to the A-loop induces its motion towards opening the binding pocket and allowing the entrance of the agonists.

The interaction of the prerequisite binding mode with the P669 amino acid side chain (first mentioned in the previous Section) was observed in the starting frame of MD_rank1_ and was diminished both upon dissociation and the penetration of BITC towards its final binding pocket. BITC in the prerequisite binding mode forms mainly polar interactions with amino acids of the A-loop and the other loop (Figure 5), such as S613, P674, T675 and Q676. Towards the final binding mode, however, BITC interacts with hydrophobic amino acids such as V678, I679 and Y680. This latter observation is strengthened by the MD_holo_ run (Figure 4), where interactions with I623 and Y662 dominate. The distances between the N atom of N615 of the A-loop and the O atom of Q676 of the opposite loop (Figure 5) appear to be good indicators for A-loop opening and closing.

## 3. Materials and Methods

### 3.1. Preparation of the Ligand Structures

Ligand conformations were obtained from their respective atomic coordinate structure files of TRPA1 receptor-ligand complexes. The ligands were modified to regain their original structure, before the covalent interaction. To JT010 [4] a chlorine was added. In the case of benzylisothiocyanate (BITC, [6]) the proper bond orders and hybridization states were restored, as R-N=C=S, the hybridization states of N and S were set to sp2 and that of C to sp. Finally, in the case of bodipy-iodoacetamide [7] an iodine was added. These modifications were carried out using the builder function of PyMol (Schrödinger, New York, NY, USA) [29]. All the ligands were energy minimized by a quantum chemistry program package, MOPAC [30,31] with PM7 parametrization [31]. Hydrogens and Gasteiger–Marsili [32] partial charges were added by OpenBabel [33]. In the case of FITTED program package [13,14,15,34], the built in preparation steps were used with default settings. For BITC the molecular mechanics force field parameters were obtained from the general AMBER force field (GAFF) [35]. The ligand was built in Maestro [36], then semi-empirical quantum mechanics optimization was performed with MOPAC [30,31] using PM7 parametrization [31], with the gradient norm set to 0.001. After energy minimization, a further force calculation step was included, the force constant matrices were positive definite. The RED-vIII.52 [37] software was used for restrained electrostatic potential (RESP) charge calculations, using RESP-A1B fitting (compatible with the AMBER99SB-ILDN force field) after ab inito geometry optimization by GAMESS [38]. Acpype [39] was used to assign atomtypes, bond and angle parameters for topology of ligand. The missing bond stretching, angle bending and torsional parameters were calculated by the antechamber [39,40] and parmchk utilities of AmberTools program package similarly as described in [41]. Torsional parameters for R-N=C=S moiety were manually added.

### 3.2. Target Preparation

The atomic coordinate file of the ligand free TRPA1 receptor was obtained from the Protein Data Bank (PDB, [42]), under the accession code 6V9W [7]. As the four chains of the target are symmetrical (homotetramer), only one chain was used to reduce computational costs. The amino acids of a chain do not interfere with the binding of the ligand to another chain. The missing atoms and residues [43] were rebuilt using SWISS MODEL [44], and energy minimized with GROMACS [45]. The convergence threshold of the steepest descent optimization was set to 10^3^ kJ mol^−1^ nm^−1^, and that of the conjugate gradient optimization to 10 kJ mol^−1^ nm^−1^. AMBER99SB-ILDN force field [35] was used for the calculation, and a position restraint at a force constant of 10^3^ kJ mol^−1^ nm^−2^ was applied on heavy atoms. The targets were further optimized by ProCESS tool of FITTED, with the original settings [13]. In the case of AutoDock (The Scripps Research Institute, La Jolla, CA, USA), the added H atoms and partial charges were kept from energy minimization.

### 3.3. Covalent Docking with FITTED

Covalent docking calculations were carried out using FITTED [13,14,15,34]. The covalent residue (C621) and adjacent basic residue (P622) were adjusted in the graphical user interface of the program. Root mean squared deviation (*RMSD*) values were calculated between the crystallographic and representative ligand conformations, if available. All other settings were used as the default of the program. In the PREPARE step of the program, the binding site interacting amino acids were identified by leaving the crystallographic ligand in the structure, which was then removed after this step. The non-covalent docking was performed similarly, with the exception, that the covalent mode of the program was switched off.

### 3.4. Prerequisite Docking with AutoDock 4.2

Prerequisite binding calculations were performed by AutoDock [25,46,47,48,49]. The number of grid points was set to 60 × 60 × 60 with a 0.375 Å grid spacing. Lamarckian genetic algorithm was used, flexibility on all active torsions was allowed on the ligands. Ten docking runs were performed for all ligands, and the resulting ligand conformations were ranked based on their calculated free energy of binding values. The binding mode with the most favorable calculated energy of binding was ranked in the lowest rank.

### 3.5. Molecular Dynamics Simulations

The apo TRPA1 and dry docked complexes of BITC were subject to a two step energy minimization, including steepest descent and conjugate gradient algorithms as described in “Target preparation”. After energy minimization, the apo and dry docked complexes were subject to 100-ns-long MD simulations. The simulation box was filled up with explicit TIP3P [50] water molecules, and to neutralize the systems, counter-ions (sodium or chloride) were added. The maximum step size of the steepest descent algorithm was 0.5 nm, and that of the conjugate gradient algorithm was 0.05 nm. The exit tolerance level of the steepest descent algorithm was set to and 10^3^ that of the conjugate gradient algorithm to 10 kJ·mol^−1^·nm^−1^. Movement of the solute Cα atoms were restrained at a force constant of 10^3^ kJmol^−1^nm^−2^, except for that of the A-loop. Calculations were performed with programs of the GROMACS [45] software package, using the AMBER99SB-ILDN [35] force field. After energy minimization, 100-ns-long NPSA MD simulation was carried out with a time step of 2 fs. Simulated annealing temperature scheme was applied as described in [22]. Simulated annealing temperature was rescaled and controlled for both solvent and solute. The temperature was gradually increased up to 323.15 K, then lowered back to 300 K in the first 20 ns, then the simulation was continued to 100 ns with constant temperature of 300 K. Pressure was coupled by the Parrinello–Rahman algorithm and a coupling time constant of 0.5 ps, compressibility of 4.5 × 10^−5^ bar^−1^ and reference pressure of 1 bar. Particle Mesh-Ewald summation was used for long range electrostatics. Van der Waals and Coulomb interactions had a cut-off at 11 Å. Coordinates were saved at regular time-intervals of 1 ps yielding 1 × 10^3^ frames. Periodic boundary conditions were treated before analysis to center whole and recovered hydrated solute structures in the box. The original protein structure served as the basis of Cα fitting.

### 3.6. Scoring

AutoDock [25] estimates the binding free energy of the ligand (Δ*G_AD_*) with Equation (1) as a scoring function.
(1)$$\Delta G_{AD}=W_{vdW}\sum_{ij}(\frac{A_{ij}}{r_{ij}^{12}}-\frac{B_{ij}}{r_{ij}^{6}})+W_{hbound}\sum_{ij}E_{\left(t\right)}(\frac{C_{ij}}{r_{ij}^{12}}-\frac{D_{ij}}{r_{ij}^{10}})+W_{elec}\sum_{ij}\frac{q_{i}q_{j}}{\varepsilon_{{\left(rij\right)}_{rij}}}+W_{sol}\sum_{ij}\left(S_{i}V_{j}+S_{j}V_{i}\right)e^{\left(-\frac{r_{ij}^{2}}{2\partial^{2}}\right)}$$

*W* terms are weighting constants calibrating to an experimentally determined set of free energies. Ligand atoms are represented by *i*, and protein atoms by *j*. A Lennard–Jones 12-6 dispersion/repulsion term, a directional 12–10 h-bonding term and a screened Coulombic electrostatic potential are included. *A* and *B* parameters are taken from the Amber force field. *E*(*t*) is a directional weight based on the angle, *t*, between the probe and the target atom. *C* and *D* parameters are assigned for well-depth calculations. The final term is a desolvation potential, *V* is the volume of the atoms surrounding a given atom and *S* is a solvation parameter for weighting [51]. *δ* is a distance weighting factor. The actual distance between the ith (ligand) and jth (target) atoms is marked with *r*.

The FITTED [13,14,15,34] scoring function estimates (Δ*G_FD_*) with the sum of various terms including the number of rotatable bonds, van der Waals and electrostatic interactions and directional H-bonding contributions as described in Equation (2).
(2)$$\Delta G_{FD}=\Delta G_{0}+0.14N_{rot}+\sum (scale\;factor)\left[\left(0.26\;U_{vdW}^{inh-prot}+0.035\;U_{elec}^{inh-prot}+0.80f_{hb}(\Delta r,\Delta \alpha )\right)\right]$$

*N_rot_* is the number of rotatable bonds, *U_vdw_* and *U_elec_* are the van der Waals and electrostatic interactions based on the AMBER94 force field. The last term is the solvation contribution to the free energy of binding. Where *f*_hb_ is the electrical field strength of hydrogen bonds, *r* is the length and α is the angle of hydrogen bonds.
(3)$$E_{ij}=\overset{N_{I}N_{L}}{\underset{ij}{\sum }}\left[\frac{A_{ij}}{r_{ij}^{12}}-\frac{B_{ij}}{r_{ij}^{6}}\right]\;A_{ij}=\varepsilon_{ij}R_{ij}^{12};B_{ij}=2\varepsilon_{ij}R_{ij}^{6};R_{ij}=R_{i}+R_{j};\varepsilon_{ij}=\sqrt{\varepsilon_{i}\varepsilon_{j}}$$
where *εi* and *εj* are the potential well depths in the equilibrium distance of atom pairs of identical types; *εij* is the potential well depth in equilibrium between the *i*th (ligand) and *j*th (target) atoms; *Rij* is the internuclear distance at equilibrium between *i*th (ligand) and *j*th (target) atoms; *Ri* and *Rj* are half equilibrium distances between *ii* and *jj* atom pairs of identical types, respectively; *rij* is the actual distance between the *i*th (ligand) and *j*th (target) atoms; *N_T_* is the number of target atoms; *N_L_* is the number of ligand atoms.

### 3.7. Ranking

The basis of the structural clustering and ranking of the docked ligand conformations was their AutoDock 4.2 binding free energy values. In the respective Tables, the serial number of ranks are represented. To create one rank [41], the ligand structure with the lowest calculated free energy of binding, and its neighboring docked ligand structures within 2 Å [52] were selected. Then new ranks were opened for the remaining structures, and clustering was repeated with the same protocol. The low serial number of a rank indicates an energetically favorable binding conformation. The actual rank (N) selected from all the ranks (M) is given in the format N/M.

*RMSD*. In all cases, the structural match of the docked (***D*** in Equation (4)) binding mode to the crystallographic reference (***C***) was expressed as a root mean squared deviation (*RMSD*) value according to Equation (1).
(4)$$RMSD=\sqrt{\frac{1}{N}\sum_{n=1}^{N}{\left|D_{n}-C_{n}\right|}^{2}}$$

In Equation (4), *N* is the number of ligand heavy atoms, ***C*** is the space vector of the nth heavy atom of the crystallographic reference ligand molecule, ***D*** is the space vector of the nth heavy atom of the calculated ligand conformation. *RMSD*_best_ is the *RMSD* value of the ligand binding mode with the lowest *RMSD*.

The distance (d) between the S atom of C621 amino acid and the ligand atom that participates in the covalent binding was also measured to check the presence of covalent bond and to estimate the degree of translation necessary to move the prerequisite binding mode into the covalent binding mode (Figure 3). The d_best_ value is the smallest distance observed.

The AA_match_ (%) is the rate of identical AAs present in two different binding pockets interacting with the ligand in a 3.5 Å cut-off distance. It is calculated by the results of Appendix A.

NHA Number of heavy atoms of the agonist counts all the atoms except for hydrogens.

EI_NHA_ Efficiency index, the calculated free energy of binding is divided by the NHA of the respective agonist. The dimension is kcal/mol.

d_covalent_ The length of the covalent bond in Å.

Rank_best_ The scoring function of the program collects the results into ranks based on their calculated free energy of binding. The lowest rank contains the best energy. The rank that contains the model with the best *RMSD* value is the Rank_best_.

## 4. Conclusions

Agonist binding to TRPA1 is a dynamic process involving structural changes of the target, first at the smaller scale of the A-loop, necessary for binding site activation, then at the whole of the target, required for channel activation. The present study identified the prerequisite binding modes of three agonists and showed how the binding of a ligand to the prerequisite site can forecast its successful docking to the final binding pocket. The prerequisite binding sites proved to be milestones on the association/dissociation pathway of the agonists, important in mechanism-based design. The present study also showed how the prerequisite binding modes affect the opening of the A-loop region, a central scene of the agonist binding mechanism. The time step measured in nanoseconds necessary for binding site activation is currently hidden from experimental methods, and only the co-operation with in silico approaches can shed light on them. Thus, amino acids identified along the dynamic binding pathway will serve as new target sites for the design of reversible binding of future agonists, beyond the well-known target of the covalent binding pocket of TRPA1.

## Figures and Tables

**Figure 1 pharmaceuticals-14-00988-f001:**
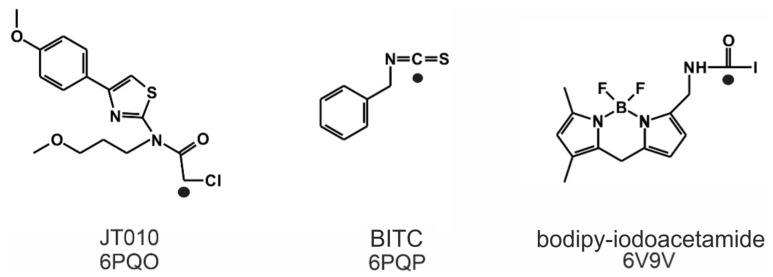
The Lewis structures of three TRPA1 agonists, JT010, benzyl-isothiocyanate (BITC) and bodipy-iodoacetamide, and the PDB IDs of their complexes with the TRPA1 receptor. The original molecular structures were restored prior to the covalent bond formation. A chlorine was added to JT010, an iodine to bodipy-iodoacetamide and the geometry of the N=C=S bond of BITC was restored. The atoms that participate in the formation of the covalent bond are marked by black dots.

**Figure 2 pharmaceuticals-14-00988-f002:**
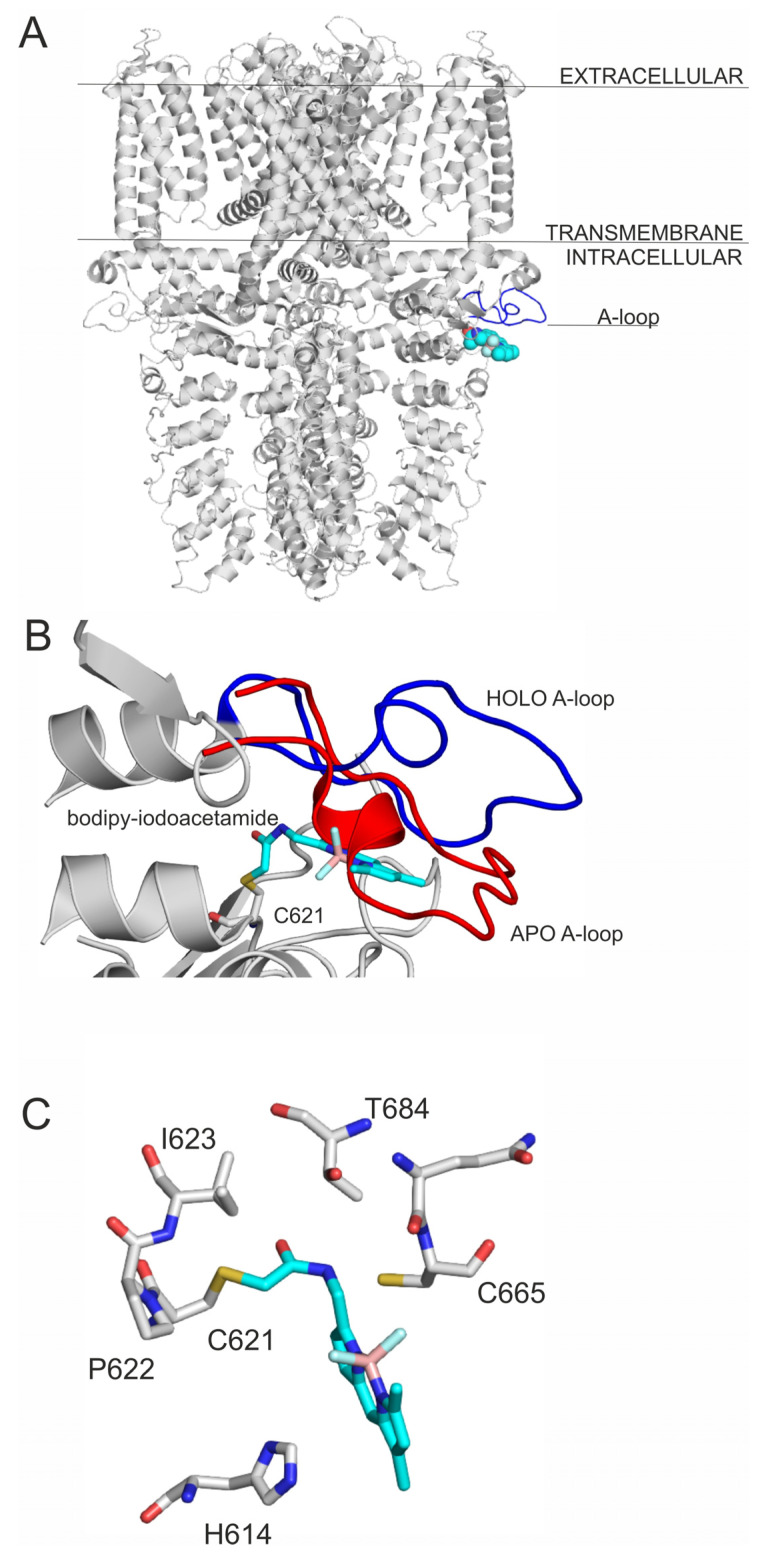
(**A**) The TRPA1 ion channel shown as grey cartoon representation, the agonist binding site is shown by the binding of bodipy-iodoacetamide (PDB: 6v9v) as teal spheres. The A-loop is highlighted with blue; (**B**) the movement of the A-loop during ligand binding. The figure was prepared with the superposition of 6v9w on 6pqo. The blue loop is A-loop in the holo, open form, and the red is A-loop in the apo, closed form. The rest of the binding site is shown with grey cartoon. bodipy-iodoacetamide is shown in teal sticks, and C621 in all atom representation grey sticks. (**C**) The close-up of the binding of bodipy-iodoacetamide to the agonist binding site of TRPA1, interacting amino acids are shown as grey thick lines in all atom representation, bodipy-iodoacetamide is shown as teal all atom representation sticks.

**Figure 3 pharmaceuticals-14-00988-f003:**
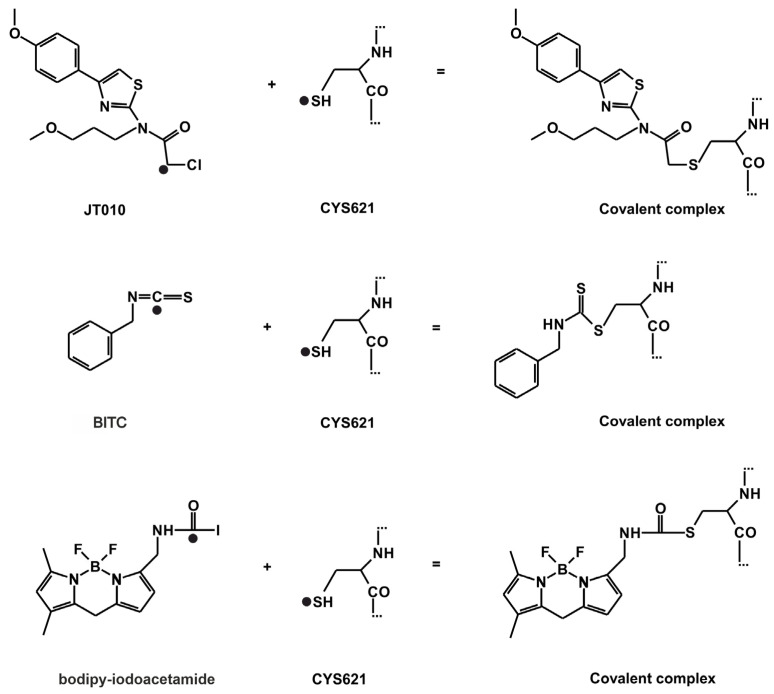
The reaction schemes of JT010, BITC, bodipy-iodoacetamide. Atoms, that participate in the formation of the covalent bond are highlighted by a black dot (●). The distance of these two atoms is referred to as d.

**Figure 4 pharmaceuticals-14-00988-f004:**
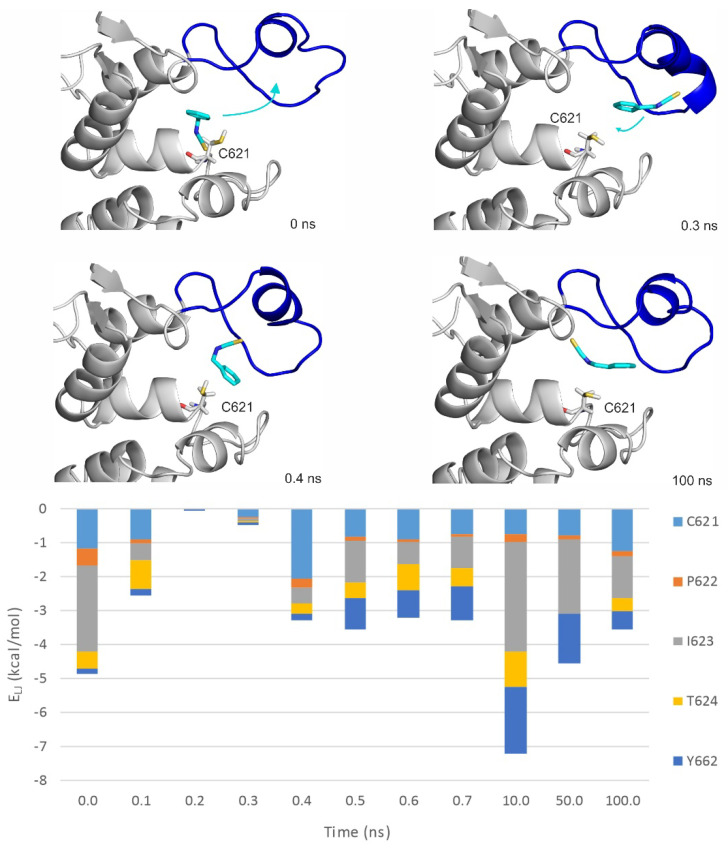
The MD_holo_ simulation starting from the experimental binding position, with the covalent bond cut. Interaction energy distribution of the interacting amino acids of the pocket (inner prerequisite site) is shown during the MD simulation. The structural figures are snapshots of the binding position of BITC at the stated time frame of the MD simulation. The protein is shown in grey cartoon and BITC with teal sticks. The A-loop is marked with blue in the open conformation. C621 amino acid is also shown as all atom sticks representation. The teal arrows indicate the movement of BITC. Lennard–Jones interaction energies calculated between BITC and the TRPA1 target amino acids are shown per residue on the diagram.

**Figure 5 pharmaceuticals-14-00988-f005:**
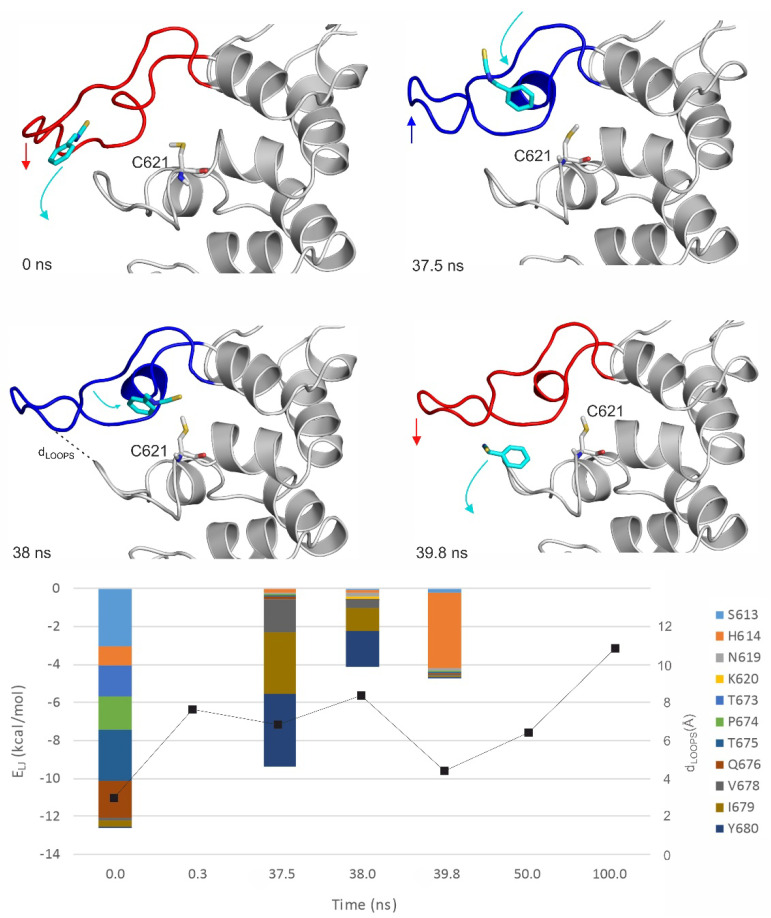
The MD_rank1_ simulation starting from the top 1st docked prerequisite binding mode of BITC on the apo TRPA1. Interaction energy distribution of the interacting amino acids of the outer prerequisite site are shown during the MD simulation. The loop motion is quantified by the distances between the N atom of N615 of the A-loop and O atom of Q676 of the opposite loop (black dotted line on the *t* = 38 ns structural plot, d_LOOPS_) shown as joint black boxes. The structural figures are snapshots of the binding position of BITC at the stated time frame of the MD simulation. The protein is shown in grey cartoon and BITC with teal sticks. The A-loop is marked with blue and red in open and closed conformations, respectively. C621 amino acid is also shown as all atom sticks representation. The red and blue arrows show the movement of the A-loop and the teal arrows the movement of BITC. Lennard–Jones interaction energies calculated between BITC and the TRPA1 target amino acids are shown per residue on the diagram.

**Table 1 pharmaceuticals-14-00988-t001:** Covalent docking calculations performed by FITTED [13,14,15].

Ligand Name	JT010	BITC	Bodipy-Iodoacetamide
HOLO target			
AA_match_ (%)	100%	100%	100%
*RMSD*_best_ (Å)	2.28	2.05	3.87
Rank_best_	1/3	1/3	1/3
Δ*G_FD_* (kcal/mol)	−84.1	−77.7	−44.3
NHA ^c^	23	10	22
EI_NHA_ ^d^ (kcal/mol)	3.66	7.77	2.01
d_covalent_ (Å)	1.8 (1.8) ^a^	1.8 (1.8) ^a^	1.8 (1.8) ^a^
APO target ^b^			
AA_match_ (%)	100%	60%	66.6%
*RMSD*_best_ (Å)	6.82	4.75	6.55
Rank_best_	1/5	1/5	1/5
Δ*G_FD_* (kcal/mol)	−77.4	−73.8	−43.1
NHA ^c^	23	10	22
EI_NHA_ ^d^ (kcal/mol)	3.36	7.38	1.96
d_covalent_ (Å)	1.8	1.8	1.8

^a^ The experimental covalent bond lengths are shown in brackets; ^b^ Without A-loop; ^c^ Number of heavy atoms; ^d^ Efficiency index.

**Table 2 pharmaceuticals-14-00988-t002:** Non-covalent docking calculations performed by FITTED.

Ligand Name	JT010	BITC	Bodipy-Iodoacetamide
HOLO target			
Δ*G_FD_* (kcal/mol)	−46.1	−32.4	−13.7
Rank_best_	10/10	1/10	8/10
AA_match_ (%)	100%	100%	100%
d_best_ (Å)	3.6	4.0	8.7
APO target ^a^			
Δ*G_FD_* (kcal/mol)	−33.4	−26.7	0.5
Rank_best_	3/5	1/5	4/5
AA_match_ (%)	100%	60%	33.3%
d_best_ (Å)	3.5	3.9	3.3

^a^ Without A-loop.

**Table 3 pharmaceuticals-14-00988-t003:** Non-covalent docking calculations performed by AutoDock 4.2 [25].

Ligand Name	JT010	BITC	Bodipy-Iodoacetamide
HOLO target			
Δ*G_AD_* (kcal/mol)	−6.8	−3.8	−5.9
Rank_best_	1/5	1/1	4/4
AA_match_ (%)	100%	80%	66%
d_best_ (Å)	3.6	6.5	4.0
APO target ^a^			
Δ*G_AD_* (kcal/mol)	−5.16	−3.74	−5.26
Rank_best_	1/3	1/2	3/5
AA_match_ (%)	50%	40%	0%
d_best_ (Å)	7.5	7.2	7.3

^a^ Without A-loop.

**Table 4 pharmaceuticals-14-00988-t004:** The details of the MD simulations performed to unravel the binding mechanism of BITC.

Simulation Name	TRPA1	Ligand	Change in A-Loop	Movement of the Agonist
MD_apo_	Apo protein	-	No change in A-loop conformation	-
MD_holo,PSA_	Holo protein	Experimental	No change in A-loop conformation	Unbinding–binding
MD_rank1_	Apo protein	Rank 1 docked ligand binding mode	A-loop flipping to the active conformation	Dissociation–association
MD_rank3_	Apo protein	Rank 3 docked ligand binding mode	No change in A-loop conformation	Dissociation

## Data Availability

Data is contained within the article and Appendix A.

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
