# Peer review of "Prerequisite Binding Modes Determine the Dynamics of Action of Covalent Agonists of Ion Channel TRPA1"

_pharmaceuticals, 2021, doi:10.3390/ph14100988_

Round 1
Reviewer 1 Report
This manuscript described the binding dynamics of TRPA1 irreversible agonists using molecular docking and dynamics methods. The authors detected prerequisite binding modes with the apo forms and showed that the reversible interactions with prerequisite binding sites contribute to structural changes of TRPA1 leading to covalent binding of the irreversible agonists. A few concerns for the authors to consider.
- The authors claimed that the irreversible agonist first interacts with the APO A-loop to open the actual binding site with C621 at the bottom. While previous publications (Neuron 2020 and Nature 2020 references) indicated that covalent binding of the irreversible agonist to C621 changed the confirmation of the agonist which including pushing out and stabilizing A-loop. There are some interactions between the irreversible agonist with the A-loop, but it is hard to determine whether these interactions (as called prerequisite interactions in the manuscript) is the prerequisite requirement for the agonist to covalently bind to C621.
- The prerequisite binding sites mentioned in this manuscript include C665, P666 and F669. Then it would be possible for the irreversible agonist to covalently modify C665. This is actually happening in real cases as indicated in previous publications. So, the modeling should go through covalent modeling instead of non-covalent modeling, which might give different results.
- In Nature 2020 reference, it is indicated that “The downward (ligand-free) A-loop,…… occluded the reactive pocket containing C621 and C665”. It looks like C665 is not that easily accessible for the agonist as described in this manuscript.
- In Neuron 2020 reference, it is demonstrated that after JT010 covalently binding to C621, “the thiazole moiety of the molecule displaces P666 and engages in interactions with Y680. This causes a large conformational change in the region containing C665 and P666”. But in this manuscript, the authors claimed that “During the transition from a non-covalent, prerequisite binding mode (d= 3.6 Å ) to the final, covalent binding mode of JT010, its interactions with F669 and Y680 diminish”.
- In Table 2 and table 3, the authors were modeling the prerequisite binding of the agonist to the A-loop. It does not make sense to exclude A-loop in the calculations.
Author Response
Reviewer 1
This manuscript described the binding dynamics of TRPA1 irreversible agonists using molecular docking and dynamics methods. The authors detected prerequisite binding modes with the apo forms and showed that the reversible interactions with prerequisite binding sites contribute to structural changes of TRPA1 leading to covalent binding of the irreversible agonists. A few concerns for the authors to consider.
- The authors claimed that the irreversible agonist first interacts with the APO A-loop to open the actual binding site with C621 at the bottom. While previous publications (Neuron 2020 and Nature 2020 references) indicated that covalent binding of the irreversible agonist to C621 changed the confirmation of the agonist which including pushing out and stabilizing A-loop. There are some interactions between the irreversible agonist with the A-loop, but it is hard to determine whether these interactions (as called prerequisite interactions in the manuscript) is the prerequisite requirement for the agonist to covalently bind to C621.
Response 1. The two above-mentioned articles indicated that the covalent binding event of an agonist to the C621 amino acid initiates great conformational changes. However, as seen in the PDB 6v9w structure the binding site is not accessible because of the overlap of A-loop with its neighbor in the apo structure (Fig. 5, the closest heavy atom distance of the two loops between N615 and D677 is 5.6 Å). Applying MD simulations we found in our manuscript, that at least a modest opening of the A-loop is necessary to allow ligand entry into the binding site cavity, and we agree, that the conformational changes on the greater scale are initiated by the final, covalent bond. Apart from interactions of the agonists with the A-loop, there are other prerequisite interactions (with amino acids of the binding pocket located between the A-loop and C621) that are necessary for the formation of the covalent bond as it was demonstrated in the results of Figs. 4 and 5. To clarify the role of prerequisite interactions a distinction of outer and inner binding modes was introduced into the main text (172-174): “As it was discussed in the Introduction, the entrance to the binding cavity (outer prerequisite binding mode) and the formation of the final ligand-target covalent bond (inner pre-requisite binding mode) is hindered by the position of the A-loop” and to the figure captions of Figs. 4 and 5. Rows 315-6 under Fig. 4: “Interaction energy distribution of the interacting amino acids of the pocket (inner prerequisite site) are shown during the MD simulation.”. Rows 326-7 under Fig. 5: “Interaction energy distribution of the interacting amino acids of the outer prerequisite site are shown during the MD simulation.”.
- The prerequisite binding sites mentioned in this manuscript include C665, P666 and F669. Then it would be possible for the irreversible agonist to covalently modify C665. This is actually happening in real cases as indicated in previous publications. So, the modeling should go through covalent modeling instead of non-covalent modeling, which might give different results.
Response 2. The Nature 2020 reference (main text 7th reference) concludes that “Other cysteines within this region (most notably C665) are also modified, but at substantially reduced rates compared to C621. Indeed, in the presence of IA we observed a weaker density associated with C665, which may reflect partial modification at this site …, as also observed with benzyl isothiocyanate.” While the above Nature 2020 reference indicates that C665 may be modified at reduced rates, still, if supposing that the above-mentioned covalent interaction with the C665 occurred, then the dissociation of the ligand (teal arrow in Fig. 5) would not be possible in reasonable time. Consequently, the downward motion of the A-loop (red arrow in Fig. 5) and its successive upward flipping and opening up of the binding pocket would not be possible according to the comparison of MD results on apo and holo systems as we concluded in Rows 288-291: “During the entire MDapo (Table 4) simulation, no significant changes were observed in the conformation of the A-loop, while, in the case of MDrank1, the loop moved upward (the red and blue arrows show the movement of A-loop and the teal arrows the movement of 9BE on Figure 5)”). In addition, there are mutational studies (7th reference of main text, Nature 2020 and 6th reference of main text, Neuron 2020) indicating that the mutation of C665 does not alter the function and activity of TRPA1 when interacting with irreversible agonists.
- In Nature 2020 reference, it is indicated that “The downward (ligand-free) A-loop,…… occluded the reactive pocket containing C621 and C665”. It looks like C665 is not that easily accessible for the agonist as described in this manuscript.
Response 3. In the apo TRPA1 structure (PDB 6v9w), the C665 is part of A-loop, and the side chain actually points outwards from the binding pocket.
- In Neuron 2020 reference, it is demonstrated that after JT010 covalently binding to C621, “the thiazole moiety of the molecule displaces P666 and engages in interactions with Y680. This causes a large conformational change in the region containing C665 and P666”. But in this manuscript, the authors claimed that “During the transition from a non-covalent, prerequisite binding mode (d= 3.6 Å ) to the final, covalent binding mode of JT010, its interactions with F669 and Y680 diminish”.
Response 4. In our top ranked docked binding mode, the thiazole S of JT010 is 5.3 Å away from the O atom of Y680, which is above our limit of 3.5 Å between two heavy atoms, to consider it an interaction. However, as the flexibility of the ligand was allowed during docking, the thiazole moiety showed rather large movements between ranks. Considering it, it is possible, that later the thiazole moiety approaches the Y680 amino acid further, which is not seen in our docking studies. In the structure of the Nature 2020 article, the thiazole S atom of JT010 is 4.9 Å away from the O atom of Y680, in the docked structure of the current manuscript the same distance is 5.3 Å. In the present manuscript we defined a strict cut-off distance of interaction of 3.5 Å for heavy atom-heavy atom distance. A clarifying note on this threshold was inserted in the manuscript (rows 233-4).
- In Table 2 and table 3, the authors were modeling the prerequisite binding of the agonist to the A-loop. It does not make sense to exclude A-loop in the calculations.
Response 5. The A-loop was included in all calculations with the holo form of TRPA1 (Tables 1-3). However, in the case of the apo TRPA1 target, the loop occludes the reactive pocket (see Point 3 above), and therefore, covalent docking of the ligands by FITTED results in steric clashes and positive binding free energies (not used in the main text). The results of FITTED apo calculations including the A-loop can be found in Tables S1 and S2. Similarly, in the case of AutoDock, non-covalent calculations were performed with (Table S3) and without the A-loop. See also the discussion of the corresponding results in the main text.
Reviewer 1 is hereby acknowledged for his/her careful work and constructive questions.
Reviewer 2 Report
The authors present a manuscript entitled “Prerequisite binding modes determine the dynamics of action of irreversible agonists of ion channel TRPA1”. The authors test the binding modes of three “potent” agonists of TRPA1 receptors, JT010, 9BE, QT4 in in silico experiments.
In general, the introduction is well written and describes the background of the channel well. The methods are described appropriately. However, I was confused about a few things, and I have some suggestions that should improve the manuscript.
My first remark actually regards the title of the manuscript: the authors write “irreversible agonists” and indeed iodoacetamide is irreversible (as mentioned by Zhao et al. 2020; PMID: 32641835). However, JT010 can be washed out in electrophysiological experiments (Suo et al. 2020, PMID: 31866091). Hence, I would suggest to exchange the term “irreversible” with “covalent” in the title.
I am also wondering about the terminology of the agonists the authors are using: While JT010 is a common name; the names of the other two agonists “9BE” and “QT4” are unusual. It took me a while to realize that these were the molecule names in the PDBs. These names cannot be found in any other paper, therefore it is impossible to find further information on them or find them in databases like the IUPHAR database. – Therefore, I suggest that the authors use the more common terminology throughout the manuscript: 9BE is called BITC (please also write the complete name), and QT4 is bodipy-iodoacetamide.
In row 71 and 72, the authors mention a new more potent compound, I assume it is JT010, which they mention in the next sentence. – the authors should mention the name JT010 in the previous sentence for clarity. For now, it might be a totally different compound.
In row 77, 9BE and QT4 were mentioned for the first time – it would be nice, if the authors could also mention the EC50 values for both compounds (as they did for JT010 in row 72). In particular, I am interested if the Bodipy-label on iodoacetamide changes its affinity for TRPA1 receptors.
The authors frequently find that QT4 (= bodipy-iodoacetamide) performs less well as the other two compounds, e.g. in row 131 or 246. – these findings are not really surprising to me, since this compound has a much lower affinity than JT010 or BITC. The authors could discuss this further.
I have a question regarding the methods and I have to mention that I am not an MD simulations specialist. While it is clear to me that only one protein chain was used for their calculations, how many molecules of the compounds were used? What I am wondering: can you account for different concentrations that are needed for binding, when agonists with vastly different affinities are used?
The authors write in their conclusion (row 333) that they used three potent agonists. Since at first I could not figure out what the real names of the compounds were, I thought the authors referred to three highly potent agonists. However, later I found out that QT4 is bodipy-iodoacetamide and assuming that it has a similar affinity as iodoacetamide, it is actually not a high-affinity compound. So maybe, the authors can clarify this.
I am really sorry, but I did not completely understand the purpose of the study: was it to show these binding modes? Or was it to test the algorithm?
If it was about the binding modes and how to use these to develop more potent agonists, then maybe the authors should emphasize more on the different binding modes by the different agonists with different affinities.
Furthermore, iodoacetamide is irreversible as opposed to JT010, despite the fact that both of them form a covalent bond – can the authors discuss this phenomenon?
Clearly, iodoacetamide reacts with any cysteine residue in any other protein. Can the authors test other channels, e.g. TRPV1 and their binding to JT010 and iodoacetamide? These channels would be a good candidate since they also react to cysteine-modification with increased, or rather sensitized, currents (Eberhardt et al. 2017, PMID: 28986540). Hence, they should possibly bind iodoacetamide but not JT010, if the complete binding pocket is necessary.
Can the authors virtually mutate TRPA1 channels? Then they could test their hypothesis by e.g. mutating F669 maybe to tyrosine and alanine to see, if this changes the binding mode. Subsequendly, one could mutate real channels and record currents and concentration response curves for wt TRPA1 channels and mutated TRPA1 channels.
Author Response
Reviewer 2
The authors present a manuscript entitled “Prerequisite binding modes determine the dynamics of action of irreversible agonists of ion channel TRPA1”. The authors test the binding modes of three “potent” agonists of TRPA1 receptors, JT010, 9BE, QT4 in in silico experiments.
In general, the introduction is well written and describes the background of the channel well. The methods are described appropriately. However, I was confused about a few things, and I have some suggestions that should improve the manuscript.
- My first remark actually regards the title of the manuscript: the authors write “irreversible agonists” and indeed iodoacetamide is irreversible (as mentioned by Zhao et al. 2020; PMID: 32641835). However, JT010 can be washed out in electrophysiological experiments (Suo et al. 2020, PMID: 31866091). Hence, I would suggest to exchange the term “irreversible” with “covalent” in the title.
Response 1. The title is changed according to the suggestion of Reviewer 2.
- I am also wondering about the terminology of the agonists the authors are using: While JT010 is a common name; the names of the other two agonists “9BE” and “QT4” are unusual. It took me a while to realize that these were the molecule names in the PDBs. These names cannot be found in any other paper, therefore it is impossible to find further information on them or find them in databases like the IUPHAR database. – Therefore, I suggest that the authors use the more common terminology throughout the manuscript: 9BE is called BITC (please also write the complete name), and QT4 is bodipy-iodoacetamide.
Response 2. In agreement with the above suggestions, the terminology was replaced to BITC and bodipy-iodoacetamide, respectively.
- In row 71 and 72, the authors mention a new more potent compound, I assume it is JT010, which they mention in the next sentence. – the authors should mention the name JT010 in the previous sentence for clarity. For now, it might be a totally different compound.
Response 3. JT010 is now included in both sentences.
- In row 77, 9BE and QT4 were mentioned for the first time – it would be nice, if the authors could also mention the EC50 values for both compounds (as they did for JT010 in row 72). In particular, I am interested if the Bodipy-label on iodoacetamide changes its affinity for TRPA1 receptors.
Response 4. Unfortunately, we could not find EC50 values even after an extensive literature research. The EC50 of allyl-isothiocyanate in activating TRPA1 is 37 nM (Nature 2020 reference). The EC50 of iodoacetamide in activating TRPA1 is 357 µM (Macpherson et al. 2007).
- The authors frequently find that QT4 (= bodipy-iodoacetamide) performs less well as the other two compounds, e.g. in row 131 or 246. – these findings are not really surprising to me, since this compound has a much lower affinity than JT010 or BITC. The authors could discuss this further.
Response 5. We could not find the EC50 value of bodipy-iodoacetamide, only the non-labelled iodoacetamide (previous question). Please, also refer to Point 9.
- I have a question regarding the methods and I have to mention that I am not an MD simulations specialist. While it is clear to me that only one protein chain was used for their calculations, how many molecules of the compounds were used? What I am wondering: can you account for different concentrations that are needed for binding, when agonists with vastly different affinities are used?
Response 6. In MD simulations, generally a single molecular entity (a target or a ligand or a target-ligand complex) is used, in the surrounding of several thousands of explicit water molecules. Ligand binding affinity (binding free energy, ΔG) can be calculated using a target-ligand complex structure as an input. For a review of such calculations, please refer to (Hollingsworth et al. 2018). In the present study, two different structure-based ΔG calculators were applied as described in Methods.
- The authors write in their conclusion (row 333) that they used three potent agonists. Since at first I could not figure out what the real names of the compounds were, I thought the authors referred to three highly potent agonists. However, later I found out that QT4 is bodipy-iodoacetamide and assuming that it has a similar affinity as iodoacetamide, it is actually not a high-affinity compound. So maybe, the authors can clarify this.
Response 7. We agree with this remark. The word “potent” is deleted from row 333. For clarity, the full names and/or abbreviations of the compounds were inserted throughout the text (Point 2).
- I am really sorry, but I did not completely understand the purpose of the study: was it to show these binding modes? Or was it to test the algorithm?
Response 8. The study includes both testing of the algorithms using known structural data (most of it was done in Section 2.1), and exploring unknown (pre-requisite) binding modes and dynamics (mostly in the next Sections). This was mentioned in the abstract as “Following a test of docking methods focused on the final, holo structures, prerequisite binding modes were detected involving the apo forms.” and also in the main text in details.
- If it was about the binding modes and how to use these to develop more potent agonists, then maybe the authors should emphasize more on the different binding modes by the different agonists with different affinities.
Response 9. According to the suggestion of the Reviewer further discussions were inserted on different binding modes by the different agonists with different affinities (Rows 253-6).
- Furthermore, iodoacetamide is irreversible as opposed to JT010, despite the fact that both of them form a covalent bond – can the authors discuss this phenomenon?
Response 10. The problem is undoubtedly interesting for discussion. However, correct answering of this question may require a quantum chemical approach and reaction path calculations which is well beyond the scope of the present manuscript.
- Clearly, iodoacetamide reacts with any cysteine residue in any other protein. Can the authors test other channels, e.g. TRPV1 and their binding to JT010 and iodoacetamide? These channels would be a good candidate since they also react to cysteine-modification with increased, or rather sensitized, currents (Eberhardt et al. 2017, PMID: 28986540). Hence, they should possibly bind iodoacetamide but not JT010, if the complete binding pocket is necessary.
Response 11. This may be a good idea for a next study focusing on TRPV1 as target. In the present case, we would restrict our discussion on TRPA1.
- Can the authors virtually mutate TRPA1 channels? Then they could test their hypothesis by e.g. mutating F669 maybe to tyrosine and alanine to see, if this changes the binding mode. Subsequendly, one could mutate real channels and record currents and concentration response curves for wt TRPA1 channels and mutated TRPA1 channels.
Response 12. According to the above suggestion of Reviewer 2 a further check was performed with virtual mutation of the TRPA1 receptor. The F669 was mutated to A669, and the binding cavity became accessible for the ligand. In the case of docking with F669 there were 3 ranks out of 10 docking runs (teal sticks). In the case of docking with A669, there was only 1 rank (orange sticks) that could go near the C621 amino acid residue inside the binding cavity. These results are also inserted into the main text rows 223-4 and into the supporting document.
Reviewer 2 is hereby acknowledged for his/her in-depth review and prospective suggestions.
Round 2
Reviewer 1 Report
The authors did answer most of my previous questions. I have a few concerns about the revised manuscript.
- The manuscript described a prerequisite binding mode for covalent agonists of ion channel TRPA1 through modeling studies. But no predictions have been provided about what kind of structures would be able to perform this prerequisite binding and further react with the Cys621 reside in the real binding pocket.
- If no predictions for further discovery of covalent agonists can be obtained, why we need this study? We know those covalent compounds do get into the binding pocket and they do interact with the A-loop from previous studies.
Author Response
Reviewer 1
The authors did answer most of my previous questions. I have a few concerns about the revised manuscript.
- The manuscript described a prerequisite binding mode for covalent agonists of ion channel TRPA1 through modeling studies. But no predictions have been provided about what kind of structures would be able to perform this prerequisite binding and further react with the Cys621 reside in the real binding pocket.
Response to question 1. As the title of the manuscript indicates, the study is focused on the exploration of prerequisite binding modes of covalent agonists and the exploration of their binding dynamics to TRPA1, and not to make ligand-based predictions. Here, our goal was to provide additional information for the target-based (mechanism-based design). We would not repeat the detailed results and discussion of the study here as they were already presented in the manuscript. Still, we would emphasize that this study provided remarkably new information for targeting TRPA1 by pointing to the importance and giving a precise account of prerequisite binding sites (target residues). In addition, unraveling binding dynamics of these compounds via the prerequisite binding sites, our study showed how the ligands actually get to the binding situation with Cys621, which is a fairly complex procedure. Thus, while ligand features were not extracted or proposed, on the target site we provided the above new information generally usable for the design of ligands of various chemistry.
- If no predictions for further discovery of covalent agonists can be obtained, why we need this study? We know those covalent compounds do get into the binding pocket and they do interact with the A-loop from previous studies.
Response to question 2. As it was mentioned in the above response to question 1, predictions were provided on the target side, and this study supplied hitherto uncovered details of the binding mechanism. Of course, we know that the covalent compounds bind to the binding pocket and the A-loop is involved in binding. The main goal of the study was to elucidate HOW all these events happen and provide additional information on the prerequisite binding events and dynamics on the target side (see Point 1 and the manuscript for details) for mechanism-based design.
The Reviewer is hereby acknowledged for his/her additional comments.
Reviewer 2 Report
The authors have answered all my questions and addressed all my concerns.
Author Response
The Reviewer is hereby acknowledged for his/her work.